# Primary Effusion Lymphoma, Multicentric Castleman’s Disease, and Kaposi’s Sarcoma in an HHV-8 and HIV-Positive Patient: A Case Report

**DOI:** 10.3390/medicina59061049

**Published:** 2023-05-30

**Authors:** Juan Carlos Caballero, Laura Pardo, Maria Socorro Rodriguez-Pinilla, Miguel Angel Piris, Beatriz Alvarez, Laura Solan, Javier Cornago, Jose Luis Lopez-Lorenzo, Pilar Llamas, Raul Cordoba, Alberto Lopez-Garcia

**Affiliations:** 1Department of Hematology, Fundacion Jimenez Diaz University Hospital, 28040 Madrid, Spain; juan.cabalero@quironsalud.es (J.C.C.); laura.pardo@quironsalud.es (L.P.); laura.solan@quironsalud.es (L.S.); javier.cornago@hospitalreyjuancarlos.es (J.C.); jllopez@quironsalud.es (J.L.L.-L.); pllamas@fjd.es (P.L.); raul.cordoba@fjd.es (R.C.); 2Department of Pathology, Fundacion Jimenez Diaz University Hospital, 28040 Madrid, Spain; smrodriguez@quironsalud.es (M.S.R.-P.); miguel.piris@quironsalud.es (M.A.P.); 3Department of Internal Medicine, Fundacion Jimenez Diaz University Hospital, 28040 Madrid, Spain; balvarez@quironsalud.es

**Keywords:** PEL, HIV, lymphoma

## Abstract

Primary effusion lymphoma (PEL), Kaposi’s sarcoma (KS), and multicentric Castleman’s disease (MCD) is an uncommon group of diseases included in the same spectrum with related characteristics. The coexistence of all of them in the same individual is a rare occurrence. We present the case of a 25-year-old patient diagnosed with human immunodeficiency virus (HIV) and the development of all these related pathologies. Despite the use of intensive treatment according to the latest recommendations, the evolution was unfavorable. This case reflects the need for new therapies and research in this field.

## 1. Introduction

PEL is a rare subtype of non-Hodgkin’s lymphoma (NHL) characterized by serous involvement of cavities, causing effusions rather than obvious masses [1]. Up to 80% of cases occur in individuals with both HIV and Epstein–Barr virus (EBV) infection, but the defining feature of PEL is an infection of the tumor cells via human herpesvirus 8 (HHV-8) [1,2]. Cell morphology can be variable in many cases with plasmacytoid differentiation and characteristically with the loss of B-cell markers such as cluster of differentiation (CD) 20 [2].

In the same spectrum of entities and closely related to HHV-8, MCD is another lymphoproliferative syndrome with plasmablastic differentiation [3].

Gold standard treatment for this lymphoma is still under debate due to its low incidence and lack of randomized studies, but it seems indisputable that the management should include intensive chemotherapy regimens [4].

Our case represents the whole spectrum of malignant pathologies characteristically associated with HIV, immunosuppression, and HHV-8 infection. Moreover, it reflects how we are still far from an effective and standardized protocol for this type of patient.

## 2. Case Presentation

We present the case of a 25-year-old male with a history of multiple sexually transmitted diseases, including syphilis, gonococcal urethritis, and type 16 human papillomavirus infection. In mid-2013, viral copies increased rapidly, reaching 600,000 VIH copies/mL and being diagnosed as stage A-I HIV infection (Figure 1). Following the 2013 guidelines and due to the elevated viral load, treatment was started with highly active antiretroviral therapy (HAART): darunavir, ritonavir, and Emtricitabine/tenofovir; decreasing viremia to undetectable levels in 2 months. Later, in 2014, violaceous macular lesions on the thorax appeared, diagnosed with a cutaneous KS with positivity for HHV-8, CD32, and CD34. Ten cycles of liposomal doxorubicin were programmed until only some isolated residual lesions remained. No radiotherapy was needed.

By 2018, the patient complained about fever, night sweats, asthenia, and pruritus. A body computerized tomography (CT) scan was performed, showing multiple adenopathies located at retroperitoneal, hiliar, and cervical regions. An excisional biopsy showed reactive lymphoid hyperplasia without HHV-8 or EBV-positive cells. With these findings, a close follow-up with routine imaging tests was performed.

In 2021, the patient reported pain in his left hypochondrium getting worse with deep inspiration. A chest X-ray was performed showing left pleural effusion (Figure 2). CT scan showed adenopathic progression, and a pleural biopsy with drainage of the effusion was also carried out. The pathology department revealed pleural tissue covered by CD138 and HHV-8-positive atypical cells with plasmablastic differentiation, concluding an infiltration by a PEL with CD20, CD3, and EBV negativity (Figure 3) [4].

To complete the lymphoproliferative syndrome diagnosis, a positron emission tomography https://en.wikipedia.org/wiki/Positron_emission_tomography (accessed on 9 May 2023) (PET)-CT scan was requested. This study showed supra and infradiaphragmatic adenopathy with fluorodeoxyglucose uptake on pleural effusion and inguinal and cervical regions (Figure 2). An excisional biopsy was taken from an inguinal node, where the same findings as described above were confirmed. In addition, HHV-8-positive vascular hyperplasia was found with multicentric LANA-1 and HHV-8-positive plasmocellular MCD morphology (Figure 4) (Table 1).

The patient was referred to the hematology department, and the case was evaluated by the committee. A regimen with bortezomib, etoposide, prednisone, vincristine, cyclophosphamide, and doxorubicin, (V-EPOCH) was decided to add valganciclovir as a treatment to reduce HHV-8 viremia. Viral loads of HHV-8 were requested in peripheral blood and were negative both at diagnosis and throughout the disease process.

After six cycles, the patient developed ascites with the persistence of the disease. Pleural effusion remained. The lymphoma committee decided to switch treatment to ifosfamide, carboplatin, etoposide regimen (ICE) plus rituximab, this last in order to target the underlying MCD.

Three cycles of treatment were completed, achieving a partial response. Ascitic effusion resolved with a slight persistence of pleural effusion. Radiotherapy was applied on the pleural affected region with the intention of obtaining the deepest response prior to a probable hematopoietic stem cell transplant (allo-HCT). To consolidate this response, an allo-HCT from a CMV-matched (negative) HLA-identical unrelated donor was performed with fludarabine, busulfan, and a total body irradiation conditioning regimen.

Despite a specific comorbidity index of 4 (high), in relation to his HIV infection and his history of KS, tolerance was acceptable with few adverse effects and no signs of graft versus host disease. The patient returned to the emergency room 40 days post-transplantation with a new pleural and ascitic effusion, confirming a new relapse after analysis and a new PET-CT scan. Due to the positivity of the neoplasm for CD38, daratumumab was requested as a drug for compassionate use, associating it with gemcitabine and oxaliplatin. As soon as the patient received this rescue therapy, 6 days later, he passed away due to multiorgan failure.

## 3. Discussion

As described above, we present a case with the full spectrum of classic diseases of HIV and HHV-8-infected patients: PEL, KS, and MCD. As described above, the patient developed malignancy despite having normal CD4 T-cell levels since HIV diagnosis.

One of the factors that contributed most to the patient’s evolution and subsequent associated diseases was the delay in initiating antiretroviral treatment [5]. Chronic antigenic stimulation and co-infection with other oncogenic viruses appear to be key factors for lymphomagenesis in HIV patients [6]. This may be the reason why even in the HAART era, the incidence of neoplasms is higher in patients living with HIV than in the general population [7]. Furthermore, an important point to highlight is the fact that in 2013 in Spain, according to GESIDA guidelines, treatment was only initiated if the patient met certain criteria, such as CD4 T-lymphocyte levels below 350 cells/μL or a viral load >100,000 copies/mL [8]. For this reason, our patient started treatment months after diagnosis, with normal CD4 levels but an uncontrolled viral load.

In parallel, the close relationship between lymphomagenesis and HHV-8 infection, even with normal CD4 lymphocyte levels, reflects the ease with which this virus alternates between its latent and lytic phases [9]. Despite having robust immunity, our patient developed all the classic complications associated with HHV-8. This ability of the virus to remain latent and/or to seek niches in the organisms that are capable of reactivating and producing disease even when immunity is optimal is still unknown to this day [10]. While the prevalence of HHV-8 ranges between 5 and 20% in Europe and the United States, the prevalence of HIV appears to be less than 0.5–1% [11]. The co-occurrence of both viruses in the same host seems to have implications for the development of related pathologies [12,13]. There are many probable reasons that led our patient to develop such an aggressive lymphoma. HIV and HHV-8 co-infection along with anthracycline exposure for his KS with consequent clonal selection were very likely the triggers [14].

A relatively new syndrome called Kaposi Sarcoma Inflammatory Cytokine Syndrome has been described in patients with KS and HIV that simulates the involvement caused by Castleman’s disease. Cytopenia, effusions, and distributive shock are some of its manifestations, usually its presentation together with immune reconstitution after the beginning of treatment with HAART [15,16]. Although, at first, this condition was considered in our patient, tumor progression seemed most likely since the control of HIV infection was optimal from the beginning and successive re-evaluation imaging tests showed a higher probability of tumor progression after successive lines of treatment.

Given the rarity of these tumors, the literature is limited, with evidence based on case series and expert opinions. In addition, the poor differentiation of these malignancies with overlapping syndromes makes it difficult to classify them into a specific category [17]. In contrast to the majority of cases in the literature, in which up to 80% of these tumors show positivity for EBER, in our case, this was negative [18,19].

As recommended by plasmacytoid differentiation lymphomas experts, we used our patient intensive first-line chemotherapy with proteasome inhibitor, bortezomib plus EPOCH. The good results observed in the larger case series were not reproduced in our patient. While most patients achieve a complete response after six cycles, our patient showed signs of progression only after three [4].

Another proposed regimen for our patient was hyper-CVAD (hyperfractionated cyclophosphamide, vincristine, doxorubicin, and dexamethasone alternating with methotrexate and cytarabine). Nevertheless, there is no data available supporting one of them [20].

As recommended by the European Group for Blood and Marrow Transplantation and according to the experience of some authors [21], autologous hematopoietic stem cell transplant (auto-HCT) should be considered in first complete remission (CR) on these patients with overall survival (OS) up to 70%. In our case, the response was not robust enough to consider it.

A recent study published by Wang et al. with patients diagnosed with HIV and Burkitt’s lymphoma shows how HIV stage does not affect the course of the hematologic disease, with other factors such as LDH at diagnosis, therapeutic regimen, and cycles received having the greatest influence on PFS and OS [22]. Though results vary depending on the used source, some studies show how HIV patients with lymphoma diagnosis have a lower OS than patients without HIV, with no impact on PFS. Moreover, in these patients, the disease tends to have more aggressive features at diagnosis [23].

A complete response has been reported on a refractory case after EPOCH and ICE with the use of the checkpoint inhibitor pembrolizumab [24]. As we could achieve a partial response after ICE, allo-HCT appeared to be the best choice. In addition, our case did not show CD19 positivity. In positive CD19 PEL, the use of axicabtagene ciloleucel (axi-cel), a CD19-directed chimeric antigen receptor (CAR) T cell therapy may be an option but is still under investigation [25].

Within the spectrum of HHV-8-positive lymphoproliferative syndromes, multicentric Castleman’s disease (MCD) takes on special relevance. Elevated IL-6 levels and HHV-8 viral load are triggers for the development of these diseases [3]. As we have shown, our case was negative for CD20. Although the consensus treatment for the second line was ICE, it was decided to add rituximab at the end of treatment to target the underlying MCD. As we have shown, our case was negative for CD20. Since the consensus treatment for the second line was ICE, it was decided to add rituximab to target the underlying MCD. With the current evidence, Rituximab should be added to patients with MCD, being able to achieve robust and durable responses [26].

The importance of the use of daratumumab in these patients should be emphasized. Recently, cases have been published with long-lasting responses in patients with PEL and regimens, including this anti-CD38 antibody [27]. A multicenter study conducted by the “*AIDS Malignancy Consortium*” has recently evaluated adding daratumumab with Dose-Adjusted EPOCH in newly diagnosed plasmablastic lymphoma [28]. Another study conducted at Columbia University in 2021 evaluates the efficacy of this approach in four patients, being able to achieve durable complete remissions of up to 17 months [29]. It is necessary to analyze these cases in order to eventually add these therapies to earlier stages of the disease since, in our case, our patient could not benefit from them.

Another special aspect of this case was the approach to suppressing HHV-8 viral load. Even though, in our case, it was not possible to quantitatively measure the patient’s viremia, there are documented reports indicating that antiviral therapy can reduce viral replication and even achieve MCD remission [30,31]. Recently, an increasing number of reports suggest the use of this antiviral in patients with these types of HHV-8-related diseases [32]. Although its use in monotherapy seems to have no clear efficacy, the latest expert recommendations suggest that its use in combination with the immunochemotherapy regimen and probably as maintenance may improve outcomes [33]. Valganciclovir was used in our patient as adjuvant therapy.

In spite of the adjuvant therapies, the outcome was ineffective. It remains unknown whether the use of these therapies prior to the onset of neoplasia could help prevent the appearance of it.

## 4. Conclusions

Due to the infrequency of this pathology, it is necessary to conduct international multicenter studies to gather more cases in order to establish a more established protocol. The use of daratumumab in the first line in CD138-positive lymphomas should probably be standardized.

## Figures and Tables

**Figure 1 medicina-59-01049-f001:**
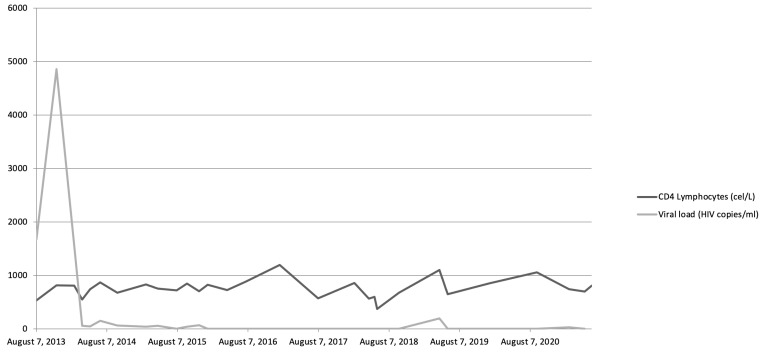
Evolution of viremia and CD4 lymphocyte levels throughout patient’s history.

**Figure 2 medicina-59-01049-f002:**
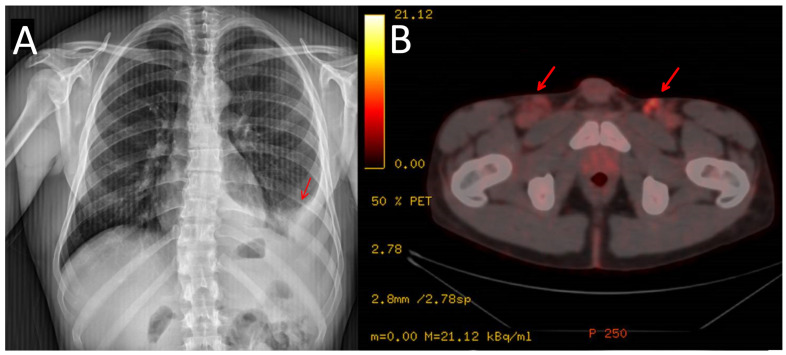
Chest X-ray showing the pleural effusion at diagnosis (**A**). PET/CT scan showing lymphadenopathies at inguinal area (marked with arrows) (**B**).

**Figure 3 medicina-59-01049-f003:**
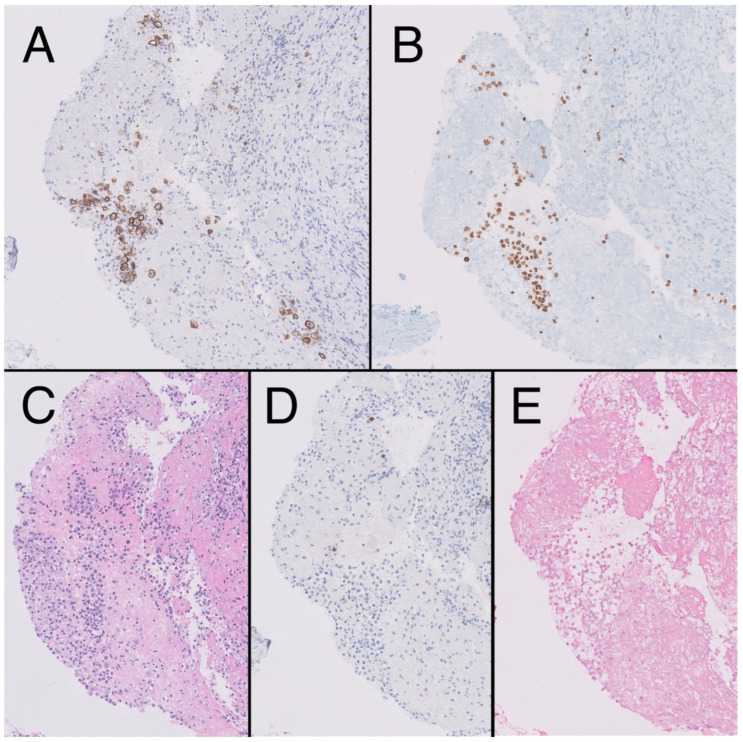
Fibroconnective tissue (10×) on pleural surface showing presence of aggregates of CD138 + atypical cells, plasmablast-like cytology (**A**), HHV8-positive (**B**), in the context of mesothelial hyperplasia, hematoxylin-eosin (HE). (**C**). Negativity for CD20 (**D**) and epstein-Barr encoding region (EBER) (**E**).

**Figure 4 medicina-59-01049-f004:**
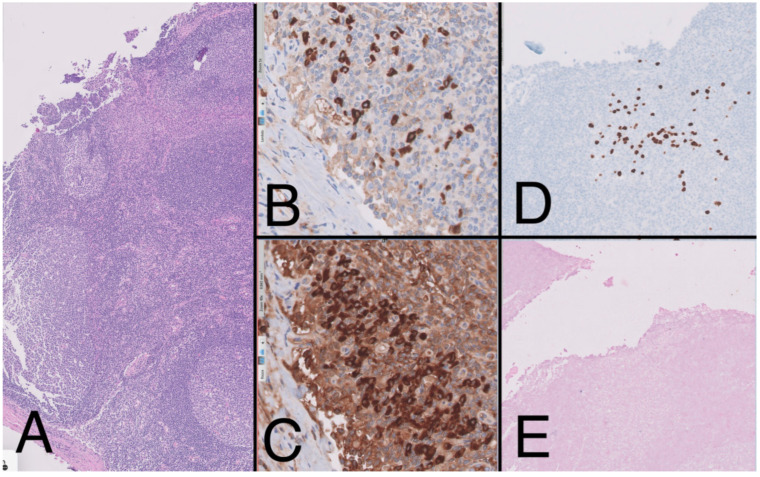
Lymph node (10×) with presence of large reactive lymphoid follicles, HE, (**A**) showing predominance of Kappa over lambda (20×) (**B**,**C**). The immunohistochemistry study (10×) shows the presence of large cells with plasmablast-like cytology and positive for HHV-8 (**D**) and negative for EBV (**E**), forming small aggregates.

**Table 1 medicina-59-01049-t001:** Disease information.

Disease Distribution	Extranodal
Viral infection	HIV, HHV-8
Positive markers	CD-138
Negative markers	CD20, CD3, EBER

## Data Availability

Not applicable.

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
