# Peer review of "Primary Effusion Lymphoma, Multicentric Castleman’s Disease, and Kaposi’s Sarcoma in an HHV-8 and HIV-Positive Patient: A Case Report"

_medicina, 2023, doi:10.3390/medicina59061049_

Round 1

Reviewer 1 Report

Caballero et al. in the present article discussed an interesting case study of the patient with HIV and HHV8 infection. As a result of co-infection patient had PEL, KS and MCD, normally a rare occurrence in a patient. Various treatment regimen prescribed showed very small or delayed response led to severe complications and resulting into death of the patient. Cases of PEL are notoriously difficult to treat and seen in HHV8 patient, there was a similar case report published by another group in 2018 (https://doi.org/10.2147/OTT.S167392). Finally, the author has suggested a need to establish protocol for the use of daratumumab as a treatment regimen for an aggressive form of tumor. There are few comments I would like to add and would like the author to explain:

1.       There was a similar case study published, author should include the reference to strengthen their observation.

2.       Daratumumab is considered a good step but should have been administered earlier.

3.       The author has not discussed any cytogenetic study or NGS based testing of the tumor?

Author Response

Response to Reviewer 1 Comments

Point 1: There was a similar case study published, author should include the reference to strengthen their observation.

Response 1: Our sincere thanks for the time you have taken to review our manuscript so thoroughly. Your reviews only make our work improve with every update. Regarding the reference you indicate, it is indeed a case with similarities with respect to ours. That is why we have included it thanks to your suggestion in our manuscript.

Point 2: Daratumumab is considered a good step but should have been administered earlier.

Response 2: Daratumumab is a very interesting therapy for this type of cases with plasmablastic differentiation and CD138 positivity. Unfortunately, its use is off-label for this type of neoplasm in our country. We have included two references in the manuscript, (lines 194-202), of some novel studies that are already evaluating the use of Daratumumab in first line.

Point 3: The author has not discussed any cytogenetic study or NGS based testing of the tumor?

Response 3: As suggested by the reviewer, we do not provide karyotype or NGS information in the text. In our center we do not routinely perform these analyses for lymphoproliferative syndromes of the type of our case. In addition, the diagnosis was clear from the beginning with flow cytometry and immunohistochemistry techniques without the need for further analysis.

Reviewer 2 Report

This is a very interesting case of likely PEL. I did not find enough information to rule out KCIS, and I would like authors to further describe this relatively new syndrome in patients co-infected with HIV and KSHV. Please provide rationale why this patient's multiorgan failure was not in fact KCIS. Detailed comments are attached here. 

Ok, minor spelling and syntax errors

Author Response

Response to Reviewer 2 Comments

Point 1: This is a very interesting case of likely PEL. I did not find enough information to rule out KCIS, and I would like authors to further describe this relatively new syndrome in patients co-infected with HIV and KSHV. Please provide rationale why this patient's multiorgan failure was not in fact KCIS. Detailed comments are attached here. 

Response 1: We would like to thank you very much for taking the time to review our article so carefully. In the text, thanks to your suggestion, we have added a small discussion indicating why we did not believe it was a KCIS. In lines 146-153 we have added two references about this syndrome. In our case the optimal control of HIV from the beginning and the fact that the neoplasm always relapsed in the same way with clear progression in the imaging tests made us rule out this syndrome.

Round 2

Reviewer 2 Report

The authors have improved the paper and I do not have any further suggestions. Congratulations! 

In my opinion paper ,can be accepted